microsystems

microelectromechanical systems, biosensor, virus detection, sialylglycopolymer, influenza

**Author for correspondence:**
Alexander S. Erofeev
e-mail: erofeev@polly.phys.msu.ru

# Label-free sensitive detection of influenza virus using PZT discs with a synthetic sialylglycopolymer receptor layer

Alexander S. Erofeev[1,2], Petr V. Gorelkin[3],
Dmitry V. Kolesov[4], Gleb A. Kiselev[5],
Evgeniy V. Dubrovin[1] and Igor V. Yaminsky[1,5]

[1]Lomonosov Moscow State University, 1, Leninskie Gory, Moscow 119991, Russia
[2]National University of Science and Technology 'MISIS', Leninskiy prospect 4, 119991 Moscow, Russia
[3]Medical Nanotechnology LLC, Skolkovo, Russia
[4]FSBSI Institute of General Pathology and Pathophysiology, 8, Baltiyskaya st., Moscow 125315, Russia
[5]Advanced Technologies Center, 4-5-47, Stroiteley Street, Moscow 119311, Russia

ASE, 0000-0002-6631-821X

We describe rapid, label-free detection of Influenza A viruses using the first radial mode of oscillations of lead zirconate titanate (PZT) piezoelectric discs with a 2 mm radius and 100 μm thickness fabricated from a piezoelectric membrane. The discs are modified with a synthetic sialylglycopolymer receptor layer, and the coated discs are inserted in a flowing virus suspension. Label-free detection of the virus is achieved by monitoring the disc radial mode resonance frequency shift. Piezo transducers with sialylglycopolymer sensor layers exhibited a long lifetime, a high sensitivity and the possibility of regeneration. We demonstrate positive, label-free detection of Influenza A viruses at concentrations below $10^5$ virus particles per millilitre. We show that label-free, selective, sensitive detection of influenza viruses by home appliances is possible in principle.

## 1. Introduction

Influenza viruses are a worldwide problem for humanity. Early detection of influenza is the most effective way to protect against the flu [1]. Many influenza transmission events are suspected to occur via aerosolized virus [2], travelling via self-contained liquid

droplets of less than 5 µm diameter [3–5]. These small droplets can remain airborne from minutes to hours and can therefore lead to a far-reaching spread of airborne infection. A recent report [6] expresses the urgent need for expedited research on influenza transmission to develop effective prevention and control strategies during an influenza epidemic. It is important to detect airborne influenza virus particles before they exceed a critical infection concentration in an indoor environment. Effective sampling of virus particles combined with highly sensitive methods of detection could enable fast, early detection of airborne influenza viruses. A successful effective approach for sampling airborne influenza virus particles that is suitable for home appliances was recently demonstrated [7]. Combining such a virus particle sampler and a sensor working in solution could be very effective for virus detection. For this, it could suffice to develop a sensor for virus detection in solution. Much more progress has been achieved in detecting virus particles in solution than in air. Sensors for influenza virus detection require several properties for use in home appliances: they should be label-free; have long lifetime durability, high sensitivity and selectivity; be insensitive to environmental conditions; and have a low cost and simple production methods.

Most similar reports use antibodies as receptors [8,9]; these methods have high selectivity but a limited lifetime and are not suitable for home appliance devices. Some groups have suggested the use of aptamers as receptor molecules [10,11]. Aptamers are synthetic analogues of antibodies with a long lifetime, but their affinity crucially depends on the spatial conformation of the aptamer molecules, which could be dramatically affected by environmental conditions. For example, temperature and pH control could be problematic in some home conditions. We previously pioneered the use of a synthetic sialylglycopolymer receptor on a cantilever transducer for sensitive, label-free, long-term virus detection [12]. We used a nanomechanical cantilever system with commercial cantilevers to measure surface stress in our previous work. A complicated optical laser system was used to measure the cantilever deflection induced by viruses binding to the receptor layer, and this approach is not suitable for home appliances due to technical issues.

Several different virus families, including Orthomyxoviridae, Paramyxoviridae and Parvoviridae, use sialoglycoconjugates for attachment [13]. Some viruses bind preferentially to sialic acids attached via a particular glycosidic linkage, and this specificity can contribute to the virus host range, tissue tropism and pathogenesis. Influenza viruses belong to the Orthomyxoviridae, which show a near obligatory dependence on the host cell surface sialoglycoprotein composition for infection. While Influenza B and C are human pathogens and use specific sialic receptors for attachment (respectively, Neu5Acα2-6Gal and Neu5,9Ac2), Influenza A viruses circulate in a wide range of avian and mammalian hosts, which determine their preferred receptor. Human Influenza A virus predominantly binds to the Neu5Acα2-6Gal sialoglycoconjugate, while avian Influenza A virus exclusively binds to Neu5Acα2-3Gal. It was recently shown that even a third saccharide in the oligosaccharide chain could also affect the affinity of the receptor to different subtypes of Influenza A viruses. Nevertheless, all species of Influenza A viruses from different hosts could bind different types of sialylglycan receptors to a varying degree, and it is hence impossible to specify an exact range of specificity [14]. Therefore, a sialoglycoconjugate receptor could be used to detect a wide range of influenza virus subtypes. Because it could also bind some other enveloped viruses, such a sensor could generate an alarm signal for further investigation of the detected pathogen. We previously demonstrated that oligosaccharides could be used as the receptor sequence to bind a certain subtype of Influenza A viruses using a stress-based mechanical sensor. We propose using a polymer matrix as the carrier for inertness and stable synthetic sialyloligosaccharide sequences. Using biologically active carbohydrate ligands is preferable, to extend the lifetime beyond the commonly used biological receptors based on natural antibodies [15–17]. It was previously shown that oligosaccharide sequences do not affect the overall receptor stability [18,19]. Using sialic saccharides instead of antibodies to detect antigenic drift of influenza virus is promising [20]. Several groups have recently developed sensors based on sialic saccharides [21,22]. Quartz crystal microbalance (QCM) and electrochemical detection systems were used in these papers and showed good sensitivity but could also be quite sensitive to non-specific massive electrically charged airborne particles.

Moreover, several types of sensors for detecting various types of avian influenza viruses were developed with different detection principles and types of readout systems, namely, fluorescent [23], electrochemical [9] and mechanical sensors [11,12].

The most promising systems for the home appliance detection of influenza virus are microelectromechanical systems (MEMS). The main advantages of MEMS sensors are the small size and simple readout system based on electronic components.

Currently, numerous publications describe the potential of using MEMS for virus detection by surface acoustic waves resonators (SAW) [8,24], a capacitive micromachined ultrasound transducer (cMUT) [25], a quartz crystal microbalance (QCM) [10,11,26,27] and a film bulk acoustic resonator (FBAR) [16,28]. Terahertz biosensing metamaterials methods [29,30] have a very high virus detection sensitivity and

could potentially be used in the future for home appliance devices, but they currently have many issues regarding selectivity in highly contaminated air with other particles with different dielectric constants and the embodiment of this technology in home appliance devices.

In our previous work [12], we described an influenza sensor with a synthetic sialylglycopolymer receptor based on a cantilever system with optical readout system. Virion adsorption to the 3′SL-PAA led to surface stress due to attractive intermolecular lateral forces in the receptor layer. MEMS have incomparable advantages due to the complex of electronic and atomic force microscopy (AFM) methods. MEMS sensor sensitivity depends not only on the effect of mass loading but also the surface stress induced in the sensor layer by binding specific target molecules [31]. Here, we describe rapid, label-free detection of Influenza A viruses using the first radial mode of lead zirconate titanate (PZT) piezoelectric discs with a 2 mm radius and 100 µm thickness fabricated from a piezoelectric membrane. Piezo discs coated with a synthetic sialylglycopolymer receptor layer are inserted in a flowing virus suspension. Label-free detection of the virus is achieved by monitoring the disc radial mode resonance frequency shift due to surface stress induced in the sensor layer by binding viruses. We demonstrate label-free, selective, sensitive detection of influenza virus particles in principle for home appliances using a well-characterized sialic acid based receptor.

# 2. Material and methods

## 2.1. Viruses

Influenza A virus A/Duck/Moscow/4182/2008 (Chumakov Federal Scientific Center for Research and Development of Immune and Biological Products of Russian Academy of Sciences) was used as our investigation target in all experiments. It contained an H5N3 surface glycoprotein composition. The stock of viruses was prepared in the allantoic cavity of 10-day-old embryonated specific-pathogen-free (SPF) chicken eggs. Allantoic liquid is a biological medium with many different proteins. This complicated mixture provides a good background for determining sensor selectivity due to total protein concentration of approximately 2 mg ml$^{-1}$ [32]. Sample solutions with concentrations of $10^5$, $10^6$ and $10^7$ virions ml$^{-1}$ were prepared by diluting the initial virus solution (approx. $1 \times 10^8$ virions ml$^{-1}$) with virus-free allantoic liquid from non-infected embryos.

## 2.2. Receptor

PZT discs were modified with synthetic sialylglycoconjugates based on a polymer matrix (3′SL-PAA) which biospecifically bind haemagglutinin proteins on the influenza viruses. Polyacrylamide conjugates bearing Neu5Acα2-3Galβ1-4Glcβ (3SL) with a high molecular weight were synthesized as previously described [33–35]. Neu5Acα2-3Galβ1-4Glcβ bearing moieties preferably binds avian subtypes of Influenza A virus and was chosen as reliable receptor for the used target object.

## 2.3. Disc fabrication and immobilization procedure

The fabrication process of PZT disc sensors from the PZT plates of 100 µm thickness with silver electrodes is presented in figure 1. We used silver-coated piezo plates (FT-21,5T-3,8AS, Quartz, Russia) for disc fabrication. Piezo plates were immersed in 98% nitric acid solution (Reahim, Russia) for 1 h to remove the silver coating. The plates were washed with Milli-Q water and ethanol several times. Subsequently, a 20 nm chromium film and 50 nm gold film were deposited by magnetron evaporation on the plate surface as an electrode and an underlying receptor layer. Plates with gold coating were incubated in a solution of 4-aminothiophenol ($10^{-3}$ M in ethanol, 16 h). We used a ground carbide-grade milling cutter as a tool to produce 4 mm discs from prepared gold-coated PZT plate. The 4 mm discs were incubated in a $10^{-3}$ M 3′SL-PAA aqueous solution overnight for better adsorption of 3′SL-PAA on the aminated gold surface. Some binding sites were deactivated after adsorption; furthermore, numerous active receptors in the polymer structure do not interact with the surface. The PZT discs were washed with Milli-Q water after 3′SL-PAA immobilization.

## 2.4. Measurement set-up

We designed a special board to measure the resonance frequency shift of PZT discs (figure 2). To measure the resonance responses of the PZT discs, we used the auto-balancing bridge method of impedance

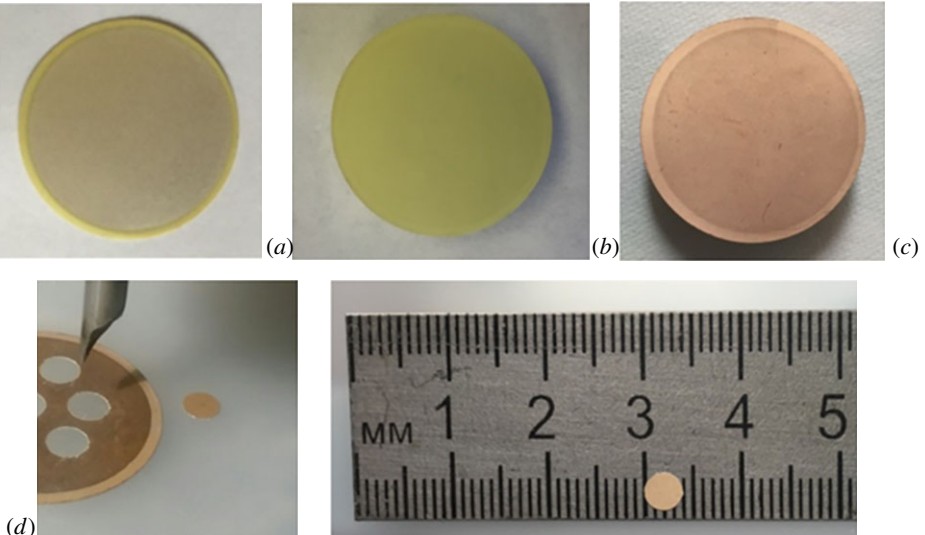

**Figure 1.** Fabrication process of piezoelectric discs: (*a*) silver-coated piezo plate with 100-μm thickness, (*b*) plate after incubation in 98% nitric acid solution, (*c*) gold- (50 nm) coated plate, and (*d*) 4 mm disc cut with chopped carbide grade from aminated gold-coated plate.

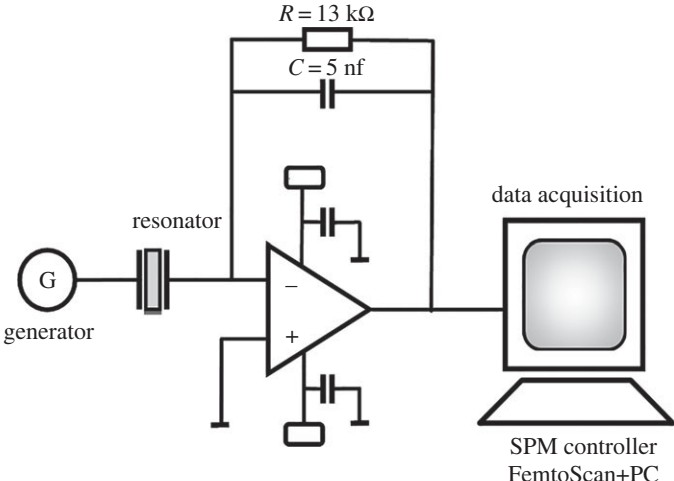

**Figure 2.** Designed board to measure resonance frequency shift of PZT discs.

measurement. We recorded the sequence of the resonance responses of the cantilever obtained with a period of 10 s. The resonant frequency was calculated according to the method of determining the 'centre of mass',

$$F_r = \frac{\sum F_i A_i}{\sum A_i},\tag{2.1}$$

where $F_r$ is the 'centre-of-mass' resonance frequency, $F_i$ and $A_i$ are the frequency and amplitude of the $i$th point of the curve. Data were provided from a FemtoScan controller for scanning probe microscopy (Advanced Technologies Center, Russia). Data processing were performed with FemtoScan Online software (Advanced Technologies Center).

In all experiments, we used PZT discs with a 2 mm radius and 100 μm thickness fabricated from a piezoelectric membrane using a common drilling machine. Each disc was clamped in the centre using a special holder with two conductive springs (figure 3*a*). The holder was placed in an Eppendorf tube, where it could be incubated in water or various solutions during experiments. This type of holding allows performing measurements at the radial mode with the shape of oscillation shown in the computer simulation model presented in figure 4.

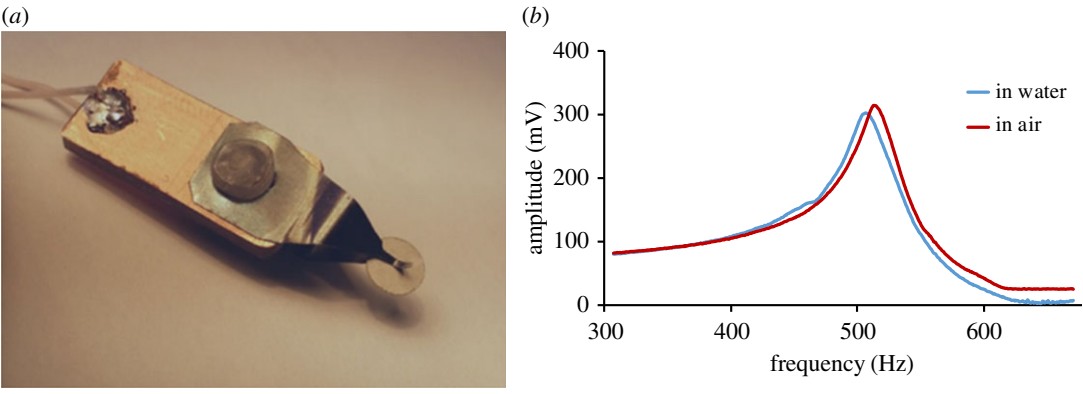

**Figure 3.** (*a*) Disc holder that clamps disc in the centre. (*b*) Resonance curves of piezo disc mechanical oscillations; red, in air; blue, in water solution.

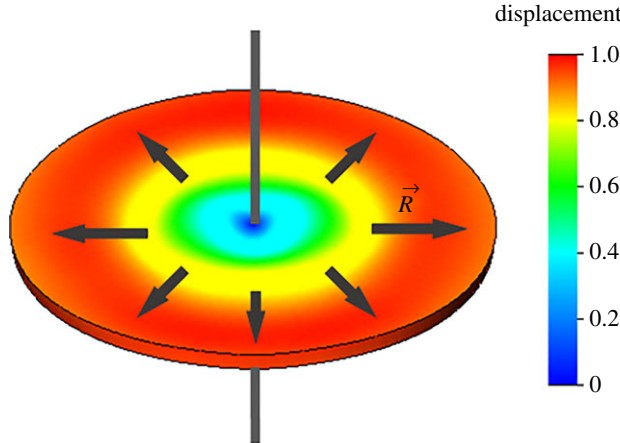

**Figure 4.** Schematic of disc first radial mode.

Using the radial mode of oscillation can exclude much of the influence of liquid on the resonance properties of the sensor. Resonance peaks of the piezo disc in water and air are presented in figure 3*b*. It was found that the Q factor changed little when air is replaced with water.

The first radial oscillation mode of a disc (figure 4) can be characterized by the equation $N_p = f_r D$, where $N_p$ is a material specific constant, $f_r$ is the eigenfrequency of first radial mode, and $D$ is the disc diameter. The eigenfrequency of the first radial mode is approximately 500 kHz for the PZT disc with 4 mm diameter, which corresponds to the experimental results in figure 3*b*.

Experiments using the piezo discs were performed with a continuous flow. A special set-up was constructed to provide measurements in a flow based on the peristaltic pump (model 77120-62, Cole-Parmer's Master Flex, Vernon Hills, USA), a two-way six-port valve (Upchurch, USA) and the liquid chamber. A peristaltic pump flow rate of 1 ml min$^{-1}$ was used for all virus detection experiments (figure 5).

## 2.5. AFM investigation

Samples for AFM investigations of a sensor layer were prepared on freshly cleaved mica using the same procedure described earlier (including Cr and Au evaporation, modification with 4-aminothiophenol and exposure to the 3'SL-PAA water solution). AFM experiments were performed in tapping mode with a Nanoscope 3a multimode atomic force microscope (Digital Instruments, USA) equipped with a tapping-mode fluid cell for operation in aqueous environments. NP-S1 silicon nitride cantilevers (Veeco, USA) with a spring constant of approximately 0.5 N m$^{-1}$ were used (working frequency in liquid was approx. 10 kHz). The line scan rate was typically 2.1 Hz, with 512 × 512 pixels per image. AFM images were processed using FemtoScan Online software (Advanced Technologies Center, Russia) [36].

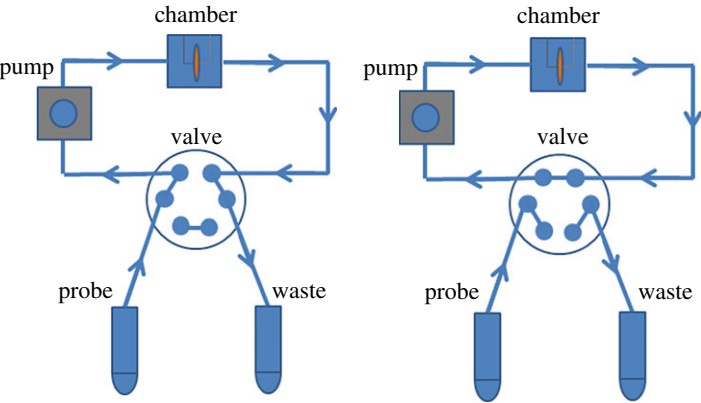

**Figure 5.** Liquid flow system consisting of camber (Eppendorf) with holder, peristaltic pump, valve with multiple positions and Teflon tubes, which provide the closed-loop system. On the left, image of valve with injection position is shown. Closed-loop system with a locked valve is shown on the right.

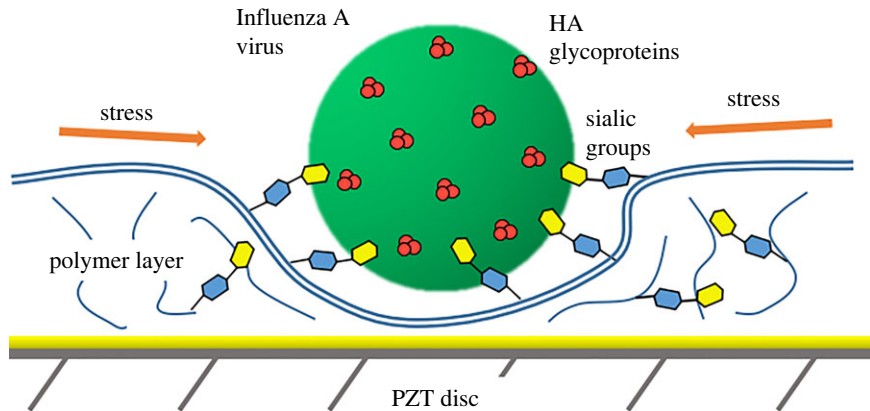

**Figure 6.** Rise of lateral stress inside the polymer receptor layer with sialyloligosaccharide groups after binding virus particle.

# 3. Results and discussion

## 3.1. Fabrication and modification of PZT disc transducers

Here, we describe label-free, rapid, easily produced sensors for influenza detection. The use of rectangular piezoceramic cantilevers for virus detection was previously demonstrated [31]. Piezoceramic cantilevers can exhibit high-frequency non-flexural resonance modes such as longitudinal extension modes due to the high piezoelectricity of the piezoelectric layer that non-piezoelectric microcantilevers such as silicon-based microcantilevers lack. Changes in the mechanical properties of the receptor layer immobilized on the piezoceramic cantilever due to the interaction with target molecules can dramatically shift the resonance peak of the longitudinal extension mode. According to Shih *et al.* [37], the relative resonance frequency shift $\Delta f/f$ is independent of the length and width of the piezoelectric cantilevers and is only proportional to the surface stress $s$ resulting from virus binding and inversely proportional to the thickness $h$ of the resonator: $\Delta f/f \propto s/h$, where $f$ is the initial resonance frequency and $\Delta f$ is the resonance frequency shift defined as the difference in the resonance frequency of the cantilever in solution before and after exposure to virus particles (figure 6).

We propose the production of sensors for virus detection based on 4 mm piezo discs. Commercially available cheap piezo plates with a 100 µm thickness were used as a PZT substrate for our resonators. Piezo discs were produced using a common drilling machine with a chopped carbide grade. This affordable method allows scalable production of disc-shaped resonators 4 mm in diameter (diameter depends on the size of the chopped cutter). We developed a special holder with central clamping (figure 3) to contact both sides of a disc. In this case, we performed experiments at the radial mode with the form of oscillation shown in figure 4. These oscillations are very similar to the longitudinal extension mode of rectangular piezo cantilevers. Using the radial mode of oscillation can exclude

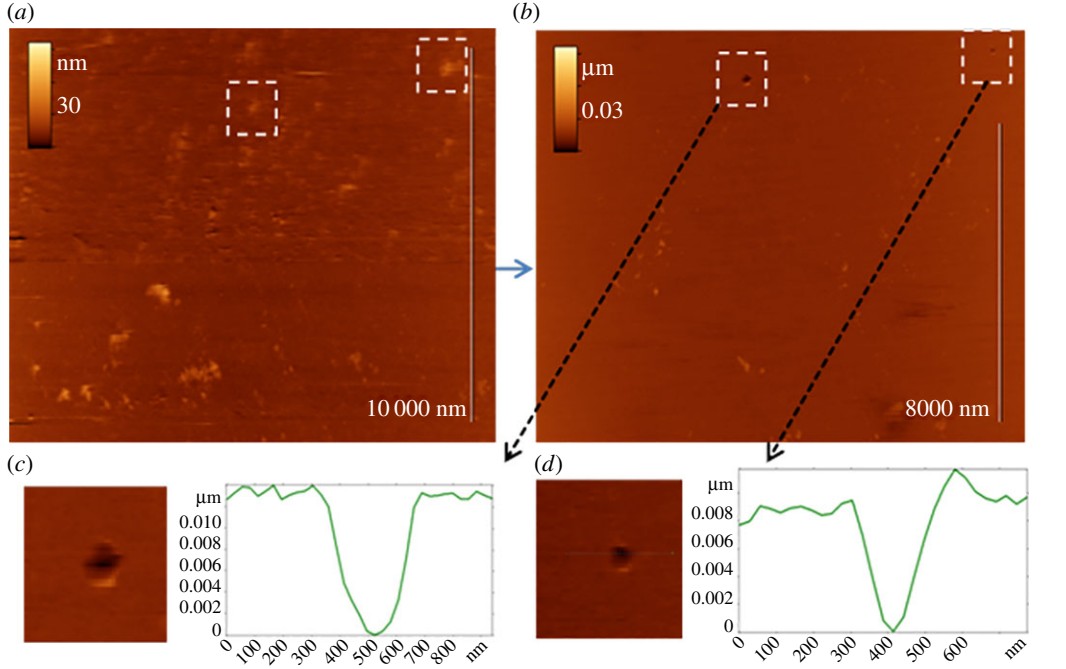

**Figure 7.** AFM images of sensor layer substrate after virus adsorption. (*a*) Virus particles are partially dipped in the receptor layer; (*b*) virus particles are removed by force and applied by AFM cantilever. Marked areas on AFM images show places with partially dipped virus before (*a*) and holes after (*b*) force application; (*c,d*) enlarged images and cross sections of remaining holes. Scan size $11 \times 14 \, \mu m^2$, insets $1 \times 1 \, \mu m^2$.

much of the influence of liquid on the resonance properties of the sensor. Resonance peaks of the piezo disc in water and air are presented in figure 3. 3′SL-PAA polymer immobilized on a piezo disc surface was chosen as the virus detection sensor layer (figure 6). Its synthetic origin provides the long lifetime of the receptor layer, which is crucial in sensors for home appliance [33–35]. Here, we propose using a polymer base with linked sialyloligosaccharides as the receptor layer on piezo discs for virus detection to increase the level of surface stress of the layer generated during biospecific binding. Polyacrylamide conjugates bearing Neu5Acα2-3Galβ1-4Glcβ are strongly attached to the gold surface of discs modified with 4-aminothiophenol. The theoretical model [38] shows that such a receptor layer should have a variable surface stress at equilibrium after absorption of viruses into the polymer layer.

Direct evidence of virus binding to the sensor surface was visualized using AFM in an aqueous environment. Figure 7*a,b* demonstrates the successive AFM images of the same area of the model biosensor surface (3′SL-PAA covered 4-aminothiophenol modified Au-Cr sputtered mica surface) after exposure to 30 μl of Influenza A virus solution ($6 \times 10^6$ vp ml$^{-1}$). The first image (figure 7*a*) demonstrates the presence of particles (globular features) 20–50 nm in height, which we identify as influenza virions bound specifically to the sensor surface. In the second image (figure 7*b*), we observe the disappearance of some particles (e.g. inside the marked areas), which can be explained by shifts due to the AFM tip during scanning. The appearance of holes (with apparent depth $\approx 10$ nm and width over 100 nm) at the positions of the viral particles after their removal (see enlarged areas in figure 7*c,d*) indicate that bound virions are partially embedded in the polymer matrix.

## 3.2. Sensitivity

Interactions between the virions and receptor layer lead to additional lateral forces [39] or additional stiffness [37], which can not only cause a negative shift in the resonance frequency of the disc, but also increase the frequency of the resonator [37]. In current experiments, the adsorption of virions from solution to the polymer receptor layer generated a positive shift in resonance frequency. We obtained meaningful differences between the frequency shifts for concentrations of 0, $10^5$, $10^6$ and $10^7$ virions ml$^{-1}$. Steady-state resonance shifts for different concentrations and the experimental dependence of the frequency shift on the concentration are presented in figure 8. The resonance frequency shift is proportional to the surface stress of the receptor layer immobilized on the sensor

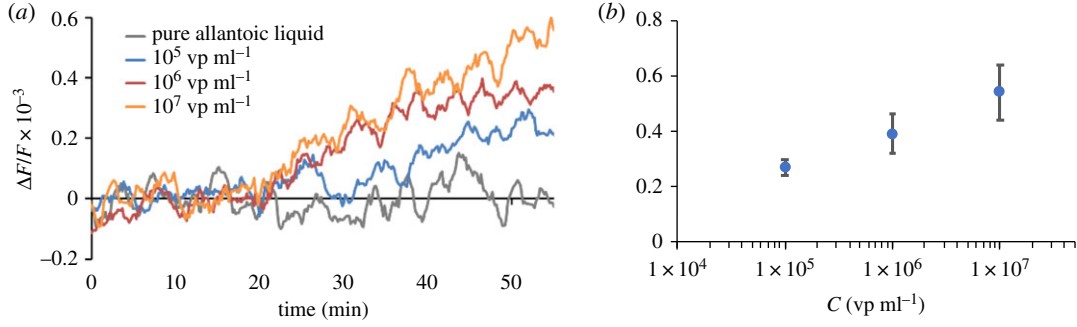

**Figure 8.** (a) Steady-state deflections of a frequency shift as a function of time for different virus concentrations (0, $10^5$, $10^6$ and $10^7$ vp $ml^{-1}$). Sample—series of virus solutions diluted with allantoic liquid. (b) Experimental dependency of frequency shift versus concentration.

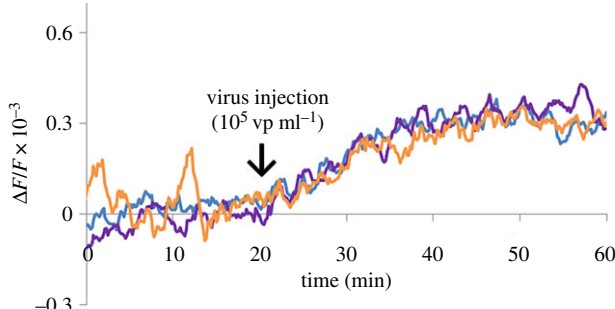

**Figure 9.** Steady-state deflections of a frequency shift as a function of time for virus concentration of $10^5$ vp $ml^{-1}$ obtained from three independent experiments.

substrate described by the Langmuir adsorption model. We recently showed that attaining the steady-state value of surface stress induced by virus adsorption takes several minutes [12,40]. Hence, the same time is required to attain the steady-state resonance shifts. Allantoic liquid containing various proteins was used as the zero probe. It did not change the receptor layer stress in contrast to solutions containing virus particles. It was confirmed that 3'SL-PAA receptors are highly specific to virions over other proteins. We obtained results with concentrations of $10^5$ virions $ml^{-1}$ and a signal-to-noise ratio of approximately 4. Sensitivity can be easily increased several fold by using a thinner PZT substrate because the sensitivity is inversely proportional to the thickness of the resonator. Applying $Pb(Mg_{1/3}Nb_{2/3})O_3-PbTiO_3$ (PMN-PT) or other materials with better electromechanical constants could also improve the sensor quality. Using piezo discs with a smaller diameter should increase the sensitivity because the density of viruses in the receptor layer would be higher. The periodic noise in all graphs could be caused by the peristaltic pump. This could be avoided by optimizing the flow system.

## 3.3. Determination of detection time and sensor reproducibility

There are three independent measurements for different (PZT) piezoelectric discs modified with a synthetic sialylglycopolymer receptor layer for the same concentration ($10^5$ vp $ml^{-1}$) in figure 9. The reproducibility of the developed sensor is confirmed by the same steady-state deflections of the frequency shift. It takes less than 20 min to exceed the plateau of resonance shift after virus injection. The proposed sensor showed acceptable detection times and reproducibility for home appliance use.

## 3.4. Regeneration possibility

In our experiments, urea (carbamide) exhibited good results for the regeneration receptor layer. We showed a regeneration possibility of urea for this receptor layer in our previous work [12]. The experiments were performed as described above. Sample solutions with $10^6$ virions $ml^{-1}$ were used to test losses of receptor layer sensitivity in regeneration experiments. Figure 10 shows the data obtained for the piezo disc sensor before (red line) and after (blue and green lines) consecutive regenerations.

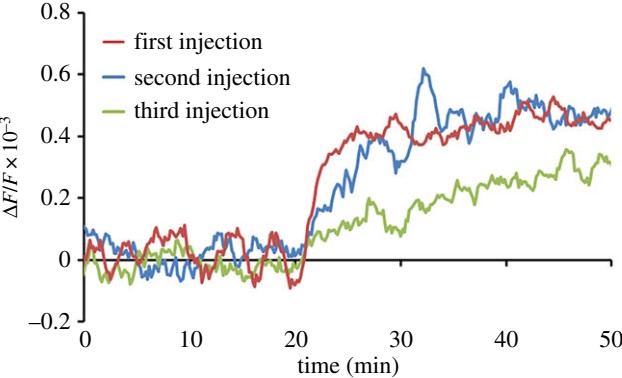

**Figure 10.** Steady-state deflections of a frequency shift as a function of time for virus concentration of $10^6$ vp ml$^{-1}$ obtained on modified gold surface by PAA-3′SL before (red line) and after (blue and green lines) consecutive regenerations.

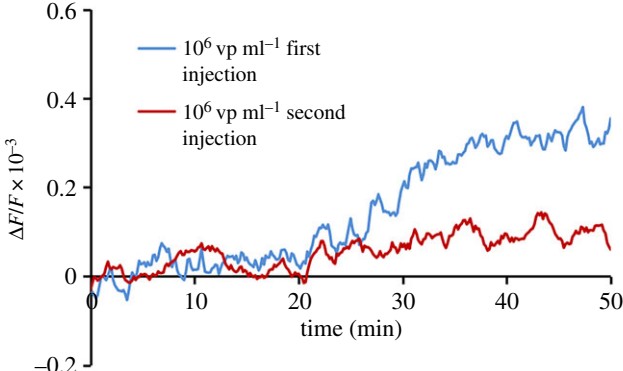

**Figure 11.** Steady-state deflections of a frequency shift as a function of time for virus concentration of $10^6$ vp ml$^{-1}$ obtained on modified gold surface by PAA-3′SL after first (blue line) and second (red line) consecutive injections.

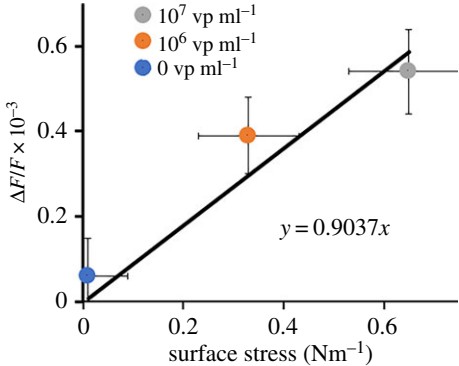

**Figure 12.** Steady-state deflections of a frequency shift as a function of surface stress obtained for virus concentrations of 0, $10^6$ and $10^7$ vp ml$^{-1}$ using a nanomechanical cantilever system. Linear interpolation of experimental dependence of $\Delta f/f \propto s$ is shown by a solid line.

Piezo disc sensor was incubated in a 10% urea water solution flow at 1 ml min$^{-1}$ for 10 min for regeneration of the receptor layer.

Less than 3% of the sensitivity was lost after the first regeneration. The equilibrium value of the resonance shift after the second regeneration was over 80% of the measurements with initial receptor layer. The fluctuation amplitude in the plateaus was equal for the experiments before and after regeneration.

Sensitivity of the sensor depends on the number of free 3′SL-PAA receptors present on the polymer layer according to the Gibbs equation relating adsorption to dependence of the surface stress [40]. The following experiment was performed to prove that the obtained resonance signals related to specific virus binding. The PZT disc with 3′SL-PAA polymer was saturated in the virus solution ($10^6$ vp ml$^{-1}$) to block

**Table 1.** Comparison of novel methods of influenza virus detection (QCM = quartz crystal microbalance, SERS + LFA = surface-enhanced Raman scattering + lateral flow assay, HA = haemagglutinin, NA = neuraminidase).

| method | monitoring use | label-free | lifetime (stability) | non-specific protection | limit of detection (LOD) | detection time | ref. |
|---|---|---|---|---|---|---|---|
| piezoceramic disc with glycopolymer receptor | yes | yes | high | high | $10^5$ vp ml$^{-1}$ for 100-μm thick below $10^4$ vp ml$^{-1}$ for 10-μm thick | 15 min | our technology |
| SERS + LFA | no | no | low | low | $1.9 \times 10^4$ PFU ml$^{-1}$ | — | [38] |
| spectrophotometry | no | no | low | high | 0.1 ng ml$^{-1}$ (approx. $1.9 \times 10^5$ vp ml$^{-1}$) | 30 min | [39] |
| QCM with aptamers in hydrogel | yes | yes | medium | low | 0.0128 HAU | 30 min | [11] |
| SAW (surface acoustic waves) | yes | yes | low | medium | 1 ng ml$^{-1}$ of HA protein | 10 min | [8] |
| LSV (linear sweep voltammetry) | yes | yes | high | high | 5.6 ng ml$^{-1}$ of NA protein poor selectivity | 30 min | [40] |

receptors. The blockage of 3′SL-PAA was controlled by measuring the resonance frequency shift (blue curve in figure 11). The sample with the virus solution ($10^6$ vp ml$^{-1}$) was injected into the system again and the resonance frequency shift was measured. The red curve in figure 11 demonstrates that the sample with viruses does not generate meaningful resonance signals of the sensor with 3′SL-PAA receptors pre-treated with virus.

## 3.5. Mechanism of action

We assumed that the relative resonance frequency shift $\Delta f/f$ is only proportional to the surface stress $s$ resulting from virus binding and inversely proportional to the thickness $h$ of the resonator: $\Delta f/f \propto s/h$. In our case, the AFM data confirmed that the thickness of the sensor layer did not change before and after virus particle interactions. In our previous paper, we measured the surface stress generation in synthetic sialylglycopolymer receptor layer [12] by virus specific sorption. We observed a proportional increase of surface stress induced by virus solutions with different concentrations. There is a nonlinear dependence of surface stress on virus concentration related to the limited number of receptor moieties on the polymer layer immobilized on the sensor substrate. The experimental dependence of $\Delta f/f \propto s$ presented in figure 12 has a good linear interpolation that confirms the origin of the resonance frequency shift by surface stress generation. We detected small non-zero changes of surface stress and $\Delta f/f$ at 0 virus particle concentration due to small non-specific interactions of proteins from the allantoic fluid.

## 3.6. Comparison with other methods

Methods for detecting influenza viruses can be divided into several groups: genetic analysis, immunoassays, electrochemical, MEMS. Laboratory and field methods can also be distinguished. Most of them are still oriented toward clinical diagnostics, but some can be used for environmental monitoring. The 'gold standard' for diagnostics are conventional PCR and ELISA, which are suitable for clinical use. Many novel methods based on improvement of traditional approaches or completely new technologies help to overcome limitations of traditional methods by increasing sensitivity, decreasing analysis time and extending the applicability. It is very complicated to compare the sensitivities of different methods because different units are used by researchers to define virus concentrations. These could be virus particles, plaque-forming units (PFUs), haemagglutination units (HAUs), molecular or weight concentrations of viruses or viral proteins. For example, for Influenza A virus particle-to-PFU ratio could be greater than 10 : 1 [41]. Table 1 presents a comparison of recent developments in influenza virus sensing on several features. We can conclude that our method shows good detection time and sensitivity as compared with others [42,43]. Several methods can show better characteristics in limit of detection (LOD) [11] or detection time [8] but are inferior in protection against non-specific binding or lifetime stability because antibodies are used. Using linear sweep voltammetry [44] for neuraminidase detection shows good stability and applicability but has poor sensitivity and selectivity due to association of NA with some bacterial infections.

# 4. Conclusion

We demonstrated the possible use of PZT piezo disc resonators with a polymer receptor layer for detecting influenza virions. The interaction between virus particles and the receptor layer on the surface of PZT discs leads to an increased frequency of the radial oscillation mode. This increase may be due to intermolecular forces in the receptor layer, which can add additional strain to the resonator system. The use of the radial oscillation mode allows excluding much of the influence of media on the resonance properties of the sensor. We detected viruses at concentrations of $10^5$ virions ml$^{-1}$ and demonstrated meaningful differences between the frequency shifts for concentrations of 0, $10^5$, $10^6$ and $10^7$ virions ml$^{-1}$. Sensitivity can be easily increased several-fold using a thinner PZT substrate, because it is inversely proportional to the thickness of the resonator. The suggested sensor satisfied the detection time and reproducibility requirements for home appliance use. We demonstrated that PZT discs with a PAA-3′SL receptor layer can be successfully regenerated. Two consecutive flow regenerations of the sensor in a 10% urea water solution led to sensitivity loss of less than 20%. We demonstrated the principle of label-free, selective, sensitive detection of Influenza A virus particles that can potentially be used to detect pathogens in home appliances.

Data accessibility. Data available from the Dryad Digital Repository: https://doi.org/10.5061/dryad.6045tk0 [45].

Authors' contributions. A.S.E., P.V.G., D.V.K., G.A.K., E.V.D. and I.V.Y. designed the research; A.S.E., P.V.G., D.V.K., G.A.K. and E.V.D. performed the experiments; A.S.E., P.V.G., D.V.K., G.A.K., E.V.D. and I.V.Y. analysed and interpreted the data; A.S.E., P.V.G., D.V.K. and E.V.D. wrote the article. All authors read and approved the final version of the article.

Competing interests. We have no competing interests.

Funding. The reported study was funded by LG Electronics under the LG-MSU Joint Laboratory, Russian Foundation for Basic Research, project no. 17-52-560001 and by the 'NUST MISiS' grant no. K4-2017-048.

Acknowledgements. The authors acknowledge Dr Alexander B. Tuzikov and Prof. Nikolay V. Bovin (The M.M. Shemyakin–Yu.A. Ovchinnikov Institute of Bioorganic Chemistry of the Russian Academy of Sciences) for assistance with polymer synthesis and Alexandra S. Gambaryan (Chumakov Federal Scientific Center for Research and Development of Immune and Biological Products of Russian Academy of Sciences) for virus preparation. The authors thank Dr Keumcheol Kwak and Dr Irina Borodina (LG Electronics) for their many helpful discussions.

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
