## [Reviewer comments · Royal Society Open Science]

Review History

RSOS-190255.R0 (Original submission)

Review form: Reviewer 1

Is the manuscript scientifically sound in its present form?

Yes

Are the interpretations and conclusions justified by the results?

No

Is the language acceptable?

Yes

Is it clear how to access all supporting data?

Yes

Do you have any ethical concerns with this paper?

No

Have you any concerns about statistical analyses in this paper?

No

Recommendation?

Major revision is needed (please make suggestions in comments)

Comments to the Author(s)

The authors applied interesting approaches to detect viruses and their experimental data suggest a potential of resonator systems as sensors for viruses. However, to confirm their claims they need to provide more experimental evidences. Please find the questions and comments as below to improve the study and manuscript.

_ There are multiple grammatical errors, so please carefully check English throughout the manuscript.

_ In the introduction the authors emphasize the need to detect virus particles in the air, but they performed detection of viruses in solution. Please rewrite the introduction section to minimize this mismatch.

_ Potential readers would not be convinced with the authors' claim that the detection method is very specific to influenza virus. It is because there is no experimental evidence for this claim. To prove their statement authors need to perform detection experiments with other enveloped viruses (for example, HIV-1, MLV, VSV, and etc.).

_ Sialic moiety can also react to other enveloped viruses. So the authors need more discussion for potential interactions between this moiety and other enveloped viruses.

_ In addition, the authors claim that the method is specific to influenza A viruses. They need to justify why other influenza virus strains will not interact with the sialic moiety.

_ Please provide more explanations on how current work is different from the authors' previous work (the 12th reference).

_ They also need to show the effect of flow rate on virus detection performance. In other word, they need to justify why the specific flow rate was chosen during detection experiment.

_ Getting physical images with AFM is not sufficient to confirm the existence of virus particles. Do you have other specific methods to prove that you actually obtained resonance signals from virus particles? One suggestion is to saturate virus particles with sialic molecules and then apply the sample onto your system. The samples should not generate resonance signals that were obtained with viruses not pre-treated with the moiety.

_ If there are molecules with sialic moiety not in excess compared with the number of virus particles, the authors should not obtain the resonance signals linearly increasing with the number of virus particles applied to the system. Please address this point by providing new experimental data or logical statement.

_ Please discuss why certain times are needed to reach state-state level of frequency in terms of physical theory about resonance.

Review form: Reviewer 2

Is the manuscript scientifically sound in its present form?

Yes

Are the interpretations and conclusions justified by the results?

Yes

Is the language acceptable?

Yes

Is it clear how to access all supporting data?

No

Do you have any ethical concerns with this paper?

No

Have you any concerns about statistical analyses in this paper?

No

Recommendation?

Major revision is needed (please make suggestions in comments)

Comments to the Author(s)

The paper titled "Label-free sensitive detection of influenza virus using PZT discs with a synthetic sialylglycopolymer receptor layer", designed and measured a biosensor for the rapid and label-free detection of Influenza A viruses. It shows the principle of label-free, selective, sensitive detection of Influenza viruses for home appliances. After a deep consideration, I think this paper can be accepted after a major revision with the following comments:

- 1) In the introduction, there are some other methods to detect the viruses, except the methods mentioned in this paper, such as the terahertz TPS, which is given in the following references:
[1] S. J. PARK, S. H. CHA, G. A. SHIN, AND Y. H. AHN, "Sensing viruses using terahertz nano-gap metamaterials", BIOMEDICAL OPTICS EXPRESS, 2017, vol.8, no.8.
[2] D Cheng, X He, X Huang, B Zhang, G Liu, G Shu, "Terahertz biosensing metamaterial absorber for virus detection based on spoof surface plasmon polaritons" International Journal of RF and Microwave Computer-Aided Engineering, 2018(7)
- 2) Much more description and explaining about fig.3 and 4 should be added in the text.
- 3) In fig.3(b), I am not very clear how much is the viurs concentration? And can you plot the curves of concentration versus frequency shift? In addition, once the viurs concentration changes, will the amplitude in Fig. 3(b) change?

Decision letter (RSOS-190255.R0)

03-Jul-2019

Dear Dr Erofeev,

The editors assigned to your paper ("Label-free sensitive detection of influenza virus using PZT discs with a synthetic sialylglycopolymer receptor layer") have now received comments from

reviewers. We would like you to revise your paper in accordance with the referee and Associate Editor suggestions which can be found below (not including confidential reports to the Editor). Please note this decision does not guarantee eventual acceptance.

Please submit a copy of your revised paper before 26-Jul-2019. Please note that the revision deadline will expire at 00.00am on this date. If we do not hear from you within this time then it will be assumed that the paper has been withdrawn. In exceptional circumstances, extensions may be possible if agreed with the Editorial Office in advance. We do not allow multiple rounds of revision so we urge you to make every effort to fully address all of the comments at this stage. If deemed necessary by the Editors, your manuscript will be sent back to one or more of the original reviewers for assessment. If the original reviewers are not available, we may invite new reviewers.

- Data accessibility

<http://datadryad.org/submit?journalID=RSOS&manu=RSOS-190255>

- Competing interests

- Authors' contributions

- Acknowledgements

- Funding statement

on behalf of Dr Derek Abbott (Associate Editor) and Pietro Cicuta (Subject Editor)
openscience@royalsociety.org

Comments to Author:

Reviewers' Comments to Author:

Reviewer: 1

Comments to the Author(s)

The authors applied interesting approaches to detect viruses and their experimental data suggest a potential of resonator systems as sensors for viruses. However, to confirm their claims they need to provide more experimental evidences. Please find the questions and comments as below to improve the study and manuscript.

_ There are multiple grammatical errors, so please carefully check English throughout the manuscript.

_ In the introduction the authors emphasize the need to detect virus particles in the air, but they performed detection of viruses in solution. Please rewrite the introduction section to minimize this mismatch.

_ Potential readers would not be convinced with the authors' claim that the detection method is very specific to influenza virus. It is because there is no experimental evidence for this claim. To prove their statement authors need to perform detection experiments with other enveloped viruses (for example, HIV-1, MLV, VSV, and etc.).

_ Sialic moiety can also react to other enveloped viruses. So the authors need more discussion for potential interactions between this moiety and other enveloped viruses.

_ In addition, the authors claim that the method is specific to influenza A viruses. They need to justify why other influenza virus strains will not interact with the sialic moiety.

_ Please provide more explanations on how current work is different from the authors' previous work (the 12th reference).

_ They also need to show the effect of flow rate on virus detection performance. In other word, they need to justify why the specific flow rate was chosen during detection experiment.

_ Getting physical images with AFM is not sufficient to confirm the existence of virus particles. Do you have other specific methods to prove that you actually obtained resonance signals from virus particles? One suggestion is to saturate virus particles with sialic molecules and then apply the sample onto your system. The samples should not generate resonance signals that were obtained with viruses not pre-treated with the moiety.

_ If there are molecules with sialic moiety not in excess compared with the number of virus particles, the authors should not obtain the resonance signals linearly increasing with the number of virus particles applied to the system. Please address this point by providing new experimental data or logical statement.

_ Please discuss why certain times are needed to reach state-state level of frequency in terms of physical theory about resonance.

Reviewer: 2

Comments to the Author(s)

The paper titled "Label-free sensitive detection of influenza virus using PZT discs with a synthetic sialylglycopolymer receptor layer", designed and measured a biosensor for the rapid and label-free detection of Influenza A viruses. It shows the principle of label-free, selective, sensitive detection of Influenza viruses for home appliances. After a deep consideration, I think this paper can be accepted after a major revision with the following comments:

- 1) In the introduction, there are some other methods to detect the viruses, except the methods mentioned in this paper, such as the terahertz TPS, which is given in the following references:
[1] S. J. PARK, S. H. CHA, G. A. SHIN, AND Y. H. AHN, "Sensing viruses using terahertz nano-gap metamaterials", BIOMEDICAL OPTICS EXPRESS, 2017, vol.8, no.8.
[2] D Cheng, X He, X Huang, B Zhang, G Liu, G Shu, "Terahertz biosensing metamaterial absorber for virus detection based on spoof surface plasmon polaritons" International Journal of RF and Microwave Computer-Aided Engineering, 2018(7)

- 2) Much more description and explaining about fig.3 and 4 should be added in the text.
- 3) In fig.3(b), I am not very clear how much is the viurs concentration? And can you plot the curves of concentration versus frequency shift? In addition, once the viurs concentration changes, will the amplitude in Fig. 3(b) change?

Author's Response to Decision Letter for (RSOS-190255.R0)

See Appendix A.

Decision letter (RSOS-190255.R1)

19-Aug-2019

Dear Dr Erofeev,

I am pleased to inform you that your manuscript entitled "Label-free sensitive detection of influenza virus using PZT discs with a synthetic sialylglycopolymer receptor layer" is now accepted for publication in Royal Society Open Science.

on behalf of Dr Derek Abbott (Associate Editor) and Pietro Cicuta (Subject Editor)
openscience@royalsociety.org

Appendix A

Response to Referees

Reviewer: 1

Remark:

The authors applied interesting approaches to detect viruses and their experimental data suggest a potential of resonator systems as sensors for viruses. However, to confirm their claims they need to provide more experimental evidences. Please find the questions and comments as below to improve the study and manuscript.

There are multiple grammatical errors, so please carefully check English throughout the manuscript.

Answer:

The grammatical errors have been corrected by our colleague whose native language is English.

Remark:

In the introduction the authors emphasize the need to detect virus particles in the air, but they performed detection of viruses in solution. Please rewrite the introduction section to minimize this mismatch.

Answer:

We rewrote the introduction and mentioned that the problem of effectively sampling airborne virus particles has already been solved. A combination of such a sampler and a sensor working in solution could be very effective for virus detection.

Remark:

Potential readers would not be convinced with the authors' claim that the detection method is very specific to influenza virus. It is because there is no experimental evidence for this claim. To prove their statement authors need to perform detection experiments with other enveloped viruses (for example, HIV-1, MLV, VSV, and etc.).

Answer:

The development of highly specific receptor layer was not a goal of our study. Sialylglycoconjugates are widely used for influenza virus detection. We used a well-known receptor molecule, which can reliably bind the chosen target avian Influenza A virus, with an innovative polymer matrix to develop a new type of micromechanical sensor based on the first radial mode resonant frequency shift. Determining the specificity of the sialyl-based receptor is a separate biochemical investigation.

Remark:

Sialic moiety can also react to other enveloped viruses. So the authors need more discussion for potential interactions between this moiety and other enveloped viruses.

Answer:

We added some discussion and a reference concerning the interaction between different types of viruses and sialic moieties in the introduction.

Remark:

In addition, the authors claim that the method is specific to influenza A viruses. They need to justify why other influenza virus strains will not interact with the sialic moiety.

Answer:

It was previously shown that the used Neu5Ac α 2-3Gal β 1-4Glc β oligosaccharide sequence preferably binds avian Influenza A viruses. Influenza C viruses exclusively bind the Neu5,9Ac2 sialic moiety, while Influenza B viruses prefer Neu5Ac α 2-6Gal. In fact, most influenza A could bind different sialic receptors with different affinity, so exact specificity could not be definitely specified [Gambaryan AS et al. Specification of receptor-binding phenotypes of influenza virus isolates from different hosts using synthetic sialylglycopolymers: non-egg-adapted human H1 and H3 influenza A and influenza B viruses share a common high binding affinity for 6'-sialyl(N-acetyllactosamine). *Virology*. 1997, 9 ,232(2), 345-50]. This is an advantage of the used receptor because it can determine the presence of a wide range of influenza pathogens, which should be further investigated to estimate the particular subtype.

We added a discussion about the receptor specificity of different types and hosts of Influenza viruses in the introduction.

Remark:

Please provide more explanations on how current work is different from the authors' previous work (the 12th reference).

Answer:

We reported completely a new type of label-free detection of viruses by monitoring the disk radial mode resonance frequency shift due to surface stress induced in the sensor layer by binding viruses. We used a nanomechanical cantilever system with commercial cantilevers to measure surface stress in our previous work. Surface stresses induced by viruses binding to the receptor layer were used as the analytic signal, which is not suitable for home appliance application because of technical issues of the detection system. We suppose that our new type of sensor potentially has all the properties needed for home appliance use for virus detection. We added this explanation in the manuscript.

Remark:

They also need to show the effect of flow rate on virus detection performance. In other word, they need to justify why the specific flow rate was chosen during detection experiment.

Answer:

We thank the reviewer for this idea. However, investigating the effect of the flow rate on virus detection performance is beyond the aim of the current work and can be a topic of separate study in the future. Though there is general understanding that the flow should renew the concentration of analyte in the area of a detector, to the best of our knowledge, detailed studies of this problem are lacking. The flow rate used in our work was in accordance with Ref. 28.

Remark:

Getting physical images with AFM is not sufficient to confirm the existence of virus particles. Do you have other specific methods to prove that you actually obtained resonance signals from virus particles? One suggestion is to saturate virus particles with sialic molecules and then apply the sample onto your system. The samples should not generate resonance signals that were obtained with viruses not pre-treated with the moiety.

Answer:

The specific response to virus particles is clearly seen from the comparison of the resonance shifts for the systems with and without viruses (Figure 8-11). AFM has been used here as a complementary method to monitor virus adhesion and for nanomanipulation. We believe that the presented results sufficiently confirm binding of the viral particles with a biosensor surface. We added experimental data confirming that resonance signals were obtained as a result of virus particle binding to the sensor layer (Figure 11). We have saturated sialic molecules immobilized at the sensor with virus particle. Hence, viruses have binded with sialic moieties in receptor layer. When we applied the sample into the system, pre-treated receptors did not catch any viral particles and meaningful resonance signals were not detected. These experimental data prove that we actually obtained resonance signals from virus particles.

Remark:

If there are molecules with sialic moiety not in excess compared with the number of virus particles, the authors should not obtain the resonance signals linearly increasing with the number of virus particles applied to the system. Please address this point by providing new experimental data or logical statement.

Answer:

We did not present any linear dependence of resonance signals on number of viruses in the previous version of the manuscript. We have plotted the dependence of resonance signals on the number of viruses in logarithmic coordinates in the current revised version of the manuscript.

Remark:

Please discuss why certain times are needed to reach state-state level of frequency in terms of physical theory about resonance.

Answer:

The resonance frequency shift is proportional to the surface stress of the receptor layer immobilized on the sensor substrate described by Langmuir and Gibbs adsorption equations [Gorelkin et al. IEEE Sensors 2015]. We recently demonstrated that the steady-state value of surface stress induced by virus adsorption takes several minutes [Gorelkin et al. Analyst 2015]. Hence, the same time is required to achieve the steady-state resonance shifts.

Reviewer: 2

Remark:

The paper titled "Label-free sensitive detection of influenza virus using PZT discs with a synthetic sialylglycopolymer receptor layer", designed and measured a biosensor for the rapid and label-free detection of Influenza A viruses. It shows the principle of label-free, selective, sensitive detection of Influenza viruses for home appliances. After a deep consideration, I think this paper can be accepted after a major revision with the following comments:

1) In the introduction, there are some other methods to detect the viruses, except the methods mentioned in this paper, such as the terahertz TPS, which is given in the following references:
[1] S. J. PARK, S. H. CHA, G. A. SHIN, AND Y. H. AHN, "Sensing viruses using terahertz nano-gap metamaterials", BIOMEDICAL OPTICS EXPRESS, 2017, vol.8, no.8.
[2] D Cheng, X He, X Huang, B Zhang, G Liu, G Shu, "Terahertz biosensing metamaterial absorber for virus detection based on spoof surface plasmon polaritons" International Journal of RF and Microwave Computer-Aided Engineering, 2018(7)

Answer:

We added information about methods with Terahertz biosensing metamaterials that have very high virus detection sensitivity and could potentially be used for home appliances in the future.

Remark:

2) Much more description and explaining about fig.3 and 4 should be added in the text.

Answer:

We added more information about in fig.3 and 4 in "2.4 Measurement setup". An additional description of fig.3 and 4 is presented in "3.1 Fabrication and modification of PZT disks transducers".

Remark:

3) In fig.3(b), I am not very clear how much is the viurs concentration?

Answer:

There were no viruses in solution and air during the resonance curve measurements of the PZT disk. Fig.3 was presented to demonstrate resonance properties of our sensor for measurements in solution.

Remark:

And can you plot the curves of concentration versus frequency shift?

Answer:

We added the experimental dependence of frequency shift versus concentration in Fig. 8b.

Remark:

In addition, once the viurs concentration changes, will the amplitude in Fig. 3(b) change?

Answer:

Virus concentration had little influence on the amplitude of the resonant curve. It can be found in the experimental raw data of resonance curves presented in the supplementary materials: <https://datadryad.org/review?doi=doi:10.5061/dryad.6045tk0>